# Estimating Vaccine Confidence Levels among Healthcare Staff and Students of a Tertiary Institution in South Africa

**DOI:** 10.3390/vaccines9111246

**Published:** 2021-10-27

**Authors:** Elizabeth O. Oduwole, Tonya M. Esterhuizen, Hassan Mahomed, Charles S. Wiysonge

**Affiliations:** 1Department of Global Health, Division of Health Systems and Public Health, Faculty of Medicine and Health Sciences, Stellenbosch University, Cape Town 7505, South Africa; hmahomed@sun.ac.za; 2Department of Global Health, Division of Epidemiology and Biostatistics, Faculty of Medicine and Health Sciences, Stellenbosch University, Cape Town 7505, South Africa; tonyae@sun.ac.za (T.M.E.); Charles.Wiysonge@mrc.ac.za (C.S.W.); 3Cochrane South Africa, South African Medical Research Council, Cape Town 7505, South Africa

**Keywords:** vaccine confidence, vaccine hesitancy, vaccination intention, immunization, COVID-19, survey, healthcare workers, South Africa

## Abstract

Healthcare workers were the first group scheduled to receive COVID-19 vaccines when they became available in South Africa. Therefore, estimating vaccine confidence levels and intention to receive COVID-19 vaccines among healthcare workers ahead of the national vaccination roll-out was imperative. We conducted an online survey from 4 February to 7 March 2021, to assess vaccine sentiments and COVID-19 vaccine intentions among healthcare staff and students at a tertiary institution in South Africa. We enrolled 1015 participants (74.7% female). Among the participants, 89.5% (confidence interval (CI) 87.2–91.4) were willing to accept a COVID-19 vaccine, 95.4% (CI 93.9–96.6) agreed that vaccines are important for them, 95.4% (CI 93.8–96.6) that vaccines are safe, 97.4% (CI 96.2–98.3) that vaccines are effective, and 96.1% (CI 94.6–97.2) that vaccines are compatible with religion. Log binomial regression revealed statistically significant positive associations between COVID-19 vaccine acceptance and the belief that vaccines are safe (relative risk (RR) 32.2, CI 4.67–221.89), effective (RR 21.4, CI 3.16–145.82), important for children (RR 3.5, CI 1.78–6.99), important for self (RR 18.5, CI 4.78–71.12), or compatible with religion (RR 2.2, CI 1.46–3.78). The vaccine confidence levels of the study respondents were highly positive. Nevertheless, this could be further enhanced by targeted interventions.

## 1. Introduction

Approximately twenty months since the appearance of the coronavirus disease 2019 (COVID-19) pandemic caused by severe acute respiratory syndrome coronavirus 2 (SARS-CoV-2) [1] on the global stage, the world is still struggling to combat and curtail the disease and its effects. More than 4 million lives of the over 187,419,838 infected by the virus have been lost across the globe as at mid-2021 [2]. The emergence of different, more virulent strains of high infectivity, of which there are at least four variants currently circulating in different parts of the world [3,4,5,6], coupled with multiple waves of the pandemic [7,8,9,10], further compound the problem. It has been suggested that in order to effectively contain the menace of the pandemic, population-wide vaccination is crucial [11,12,13]. It is widely acknowledged that the success of any vaccination endeavor depends largely on the healthcare workforce, who usually are the most trusted source of health information for the public [14,15,16]. These healthcare workers were the first to receive a COVID-19 vaccine when one became available [17] and are not immune to vaccination concerns, as previously documented [14,15,16,18,19]. The accelerated development, production, and pre-licensure emergency use authorization of the successful vaccine candidates has added pressure to the already strained public confidence in vaccines [11,20]. These may also have inadvertently fueled the vaccination misinformation on web-based information and social media platforms [21,22,23,24,25,26]. It is against this backdrop of heightened vaccine confidence deficit that healthcare workers are expected to receive, promote, and administer successful COVID-19 vaccine candidates to a pandemic-stressed, vaccine-wary public.

Moreover, healthcare workers do not operate in a vacuum; they are usually nested within healthcare systems and communities. The resilience of many healthcare systems has been severely tested since the outbreak of the pandemic [27], and the fragile ones, especially those in Africa, have struggled in the current crises [27]. This engenders a situation in which concerned healthcare workers operating within beleaguered healthcare systems have the responsibility of administering and promoting COVID-19 vaccines and vaccination to the general public. It is, however, notable that a significant measure of success has been achieved as millions of doses of the vaccines have been administered among healthcare workers and the general public alike in spite of these challenges.

These issues of concern and challenges highlighted above, amongst other reasons previously identified [20], informed the investigation of vaccine confidence levels in a cohort of healthcare workers and their trainers ahead of the COVID-19 vaccination roll-out in South Africa.

This study builds on a previous international study that examined the perceptions of vaccine importance, safety, effectiveness, and religious compatibility among 65,819 individuals across 67 countries, of which South Africa was one [28]. We expand on this study by presenting recent data and findings from a subset of the South African population.

## 2. Materials and Methods

### 2.1. Study Design

This was a cross-sectional survey targeting all academic staff and students of the Faculty of Medicine and Health Sciences at Stellenbosch University in Cape Town, South Africa. Academic staff, defined as staff that were engaged in research or teaching at undergraduate and/or postgraduate levels, were the original target of the survey, as indicated in the published protocol [20]; however, a protocol deviation led to the inclusion of non-teaching, administrative staff.

Participants were informed of their rights to participate or otherwise in the survey invite email; voluntary participation was deemed as implied consent.

### 2.2. Study Population Description and Sampling Strategy

The target population included 4304 students, of which 3016 (70.1%) were females and 1287 (30%) males; 1 (0.02%) individual identified as non-binary. This student population consisted of 1599 (37.1%) postgraduates, 2650 (61.6%) undergraduates, and 55 (1.3%) occasional students (that is, students taking modules for non-degree purposes at the university). There were 1278 staff members in the target population, of which 893 (70.0%) were females and 385 (30.0%) were males. These numbers were obtained from the institutions’ information holders.

No sampling strategy was applied as a census of the entire source population was intended. Nevertheless, the promise of a short survey completion time (the survey took an average of 3 min) and an incentive of 3 randomly selected participants with complete survey responses winning a ZAR 500 (approximately GBP 25.60) cash prize were aimed at maximizing the response rate.

### 2.3. Sample Size and Response Rate

The estimated sample size based on a finite population of between 3500 and 5000 potential participants was calculated to be between 1009 and 1105; the assumptions underpinning this calculation have been described previously [20].

### 2.4. Data Collection

The survey was conducted online using the Checkbox^®^ survey software on the Stellenbosch University Survey platform to capture participants’ responses. Data collection was conducted between the 4th of February and the 7th of March 2021, prior to the implementation of the COVID-19 vaccine roll-out. The tool of data collection was an online-administered succinct questionnaire consisting of 6 demographic questions, 5 vaccine confidence statements, and 1 intention to receive a COVID-19 vaccine when one becomes available statement (i.e., “I will take a COVID-19 vaccine when one becomes available”). The demographic information elicited included age, sex, religion, academic status (i.e., staff, student, or both), highest degree obtained, and number of years of schooling post-high school. These demographic variables are some of those that had been investigated in literature for the effect that they may have on the confidence levels of healthcare workers. Four of the 6 statements used to explore vaccine sentiments were obtained directly from the previously validated vaccine confidence statements used in a study conducted in 67 countries [28,29]. The 4 statements were: “vaccines are important for children to have”; “overall, I think vaccines are safe”; “overall, I think vaccines are effective”; and “vaccines are compatible with my religious beliefs”. The 2 newly added statements were: “vaccines are important for me to have”, and “I will take a COVID-19 vaccine when one becomes available”. The former explored the individual’s sentiments about the importance of vaccination for oneself, while the latter explored the individuals’ intention to vaccinate with particular reference to the (then) imminent COVID-19 vaccination roll-out. The process of development and the rationale for the inclusion of the two additional questions have also been previously described [20].

The full questionnaire was pilot-tested on a small sample of respondents who shared similar characteristics with the study population but were not included in the study. This was to explore the face validity, feasibility, and the logistics of administering the survey. The survey was deemed satisfactory in these three aspects. Each study participant was asked to rate the degree with which he or she agreed with the 5 vaccine confidence statements and the one statement on intention to receive a COVID-19 vaccine (i.e., “I will take a COVID-19 vaccine when one becomes available”) on a 5-point Likert scale. The scale consisted of the following ordinal categories: strongly agree, tend to agree, do not know, tend to disagree, strongly disagree. The questionnaire was in English.

### 2.5. Data Analysis

The complete responses were exported from the data collection platform Checkbox^®^ survey via Stellenbosch University surveys to a Microsoft Excel spreadsheet, where initial data cleaning and coding was done. The data set was subsequently exported into IBM^®^ SPSS Statistics software version 27 (IBM Corp. Released 2020. IBM SPSS Statistics for Windows, Version 27.0. Armonk, NY, USA: IBM Corp), where further cleaning, labelling, and analysis were performed. EpiCalc 2000 version 1.02 (Joe Gilman and Mark Myatt, Brixton Books, 1998) was used in the calculation of 95% confidence intervals, while Stata/MP version 17.0 for Windows (Stata Corp LLC, College Station, TX, USA) software was used to compute the crude relative risks using log binomial models.

The main outcome investigated was the variability in the vaccine sentiments and vaccine intention within and across all groups investigated. This was done by considering the fraction of respondents that either agreed or disagreed with the five statements on immunization and the one statement on intention to receive a COVID-19 vaccine (i.e., “I will take a COVID-19 vaccine when one becomes available”). The “strongly agree” and “tend to agree” responses were combined to make up the positive vaccine sentiment variable, while the “strongly disagree” and “tend to disagree” responses were combined to make up the negative vaccine sentiment variable. The “don’t know” responses were removed from the data prior to analyses according to the study’s methodology as previously described [20].

Categorical variables were summarized using frequencies and proportions. Quantitative variables are presented using medians and interquartile ranges as they were not normally distributed. Statistical significance was defined at a *p*-value < 0.05.

Pearson’s chi square exact 2-sided *p*-values were used to assess the association between categorical variables, while the Mann–Whitney U test was used for the quantitative variables.

The overall vaccine acceptance was >10%; therefore, a log binomial regression model was used to assess the association between vaccine confidence and the outcome of the intention to receive a COVID-19 vaccine when one becomes available, as previously indicated [20]. The strength of this association was assessed using the crude relative risk (RR), and its corresponding 95% confidence interval (CI) was calculated for each vaccine confidence statement and the intention to receive a COVID-19 vaccine. Strong internal consistency was detected between the 5 vaccine confidence statements, resulting in high levels of multicollinearity between the independent variables. Due to these high levels of multicollinearity, a multivariable model was not possible.

## 3. Results

The total number of estimated potential respondents based on the number of email invites sent out was 4659 (3737 students and 922 staff). A total of 1414 participants responded to the survey invite, giving a total response rate of 30.35%. Of the 1414 responses received, 1015 were complete responses. This gives an actual response rate of 21.79%.

The age of the 1015 respondents that submitted complete survey responses (study sample population) ranged from 16 to 90 years, with a median of 53 years. An overview of the characteristics of the study population is provided in Table 1.

The total complete responses for the five vaccine confidence statements and the intention to receive a vaccine COVID-19 vaccine are shown in Table 2. Strong internal consistency existed within the five vaccine confidence statements, as previously mentioned. Responses to all five independent variables (the five vaccine confidence statements) were highly associated with each other. Chi square tests between each pair of independent variables showed a highly statistically significant association in each case (*p* < 0.001 in all pairwise cross-tabulations). This indicated high levels of consistency in the answers to each of the five statements, and, therefore, high levels of multicollinearity. This strong internal consistency, also reflected by the Cronbach’s alpha of 0.840, made a multivariate model unviable.

Overall, the study participants showed high level of agreement with the five vaccine confidence statements. There was good precision in the estimates, of ±3%. However, the percentage of people who intended to receive a COVID-19 vaccine was slightly lower, at 89.5% (95% CI 87.19–91.38) as shown in Table 3.

The results of the associations between vaccine confidence and intention are shown in Table 4. Participants who expressed positive vaccine sentiments for vaccine safety were 32 times (95% CI 4.67–221.89) more likely to agree to receive a COVID-19 vaccine. A similar pattern was observed for the other vaccine sentiments. However, the compatibility of vaccine with religion statement showed that the likelihood of those participants who agreed that vaccines were compatible with their religious belief to indicate their willingness to receive a COVID-19 vaccine to be 2.2. Nevertheless, the effect size had comparatively higher precision at 95% CI of 1.46 to 3.78. This trend was also observed with the “vaccines are important for children to have” statement, which had a slightly higher relative risk of 3.5 (95% CI 1.78–6.99). The *p*-value for the association between each of the five vaccine confidence statements and the intention to receive a COVID-19 vaccine was *p* < 0.001, as shown in Table 4.

For categorical demographic variables, no potential predictors were found associated with either the vaccine confidence statements or the intention to receive a COVID-19 vaccine statement at the bivariate level of analysis, as shown in Appendix A.

The distributions of quantitative variables between those who agreed with and those who disagreed with the vaccine confidence statements and with the intention to receive a COVID-19 vaccine when one became available were not different between the groups, as shown in the Appendix A. Furthermore, there was no significant statistical association between any quantitative variables and vaccine confidence.

Overall, and within all demographic categories, the vaccine sentiment of the study respondents was highly positive. Most categories of respondents reported over 90% agreement with the five vaccine confidence statements. The exception (a lower than 90% level of agreement) was the 7th Day Adventist group in the religion category (7 individuals), which had 66.7% agreement with vaccine confidence statements 2 and 3 (“vaccines are important for me to have” and “overall, I think vaccines are safe”); this is shown in Appendix A. They also had 77.8% agreement with vaccine confidence statement 4 (“overall, I think vaccines are effective”), as reflected in Appendix A. However, this same category of religion had 100% agreement that vaccines were compatible with their religious beliefs (vaccine confidence statement 5), as shown in Appendix A, and that vaccines were important for children to have (vaccine confidence statement 1), as reflected in Appendix A. Any trends for this group should be interpreted with caution because of the small numbers involved.

Positive vaccine sentiments were comparably slightly less across all categorical variables for the intention to receive a COVID-19 vaccine when one becomes available than it was for the five vaccine confidence statements. Appendix A shows this with many of the factors having >80%, but mostly less than 90% agreement with the COVID-19 vaccine receipt intention. Again, the exception was the 7th Day Adventist group (only 7 persons), which had only 42.9% agreement with the statement for the receipt of a COVID-19 vaccine when one becomes available. This result is shown in Appendix A.

Due to the low numbers per cells and the many groups within the religion category, the *p*-values for vaccine confidence statements 2 and 3, and the intention to receive a COVID-19 vaccine when one becomes available statement, could not be computed. This can be seen in Appendix A. Nevertheless, there is negligible evidence to suggest that if these *p*-values had been computed, they would have been different from the other *p*-values obtained for the religion category for the other vaccine confidence statements. An example of this is shown in Appendix A, where the *p*-value for the religion category is 0.851.

No significant association between levels of education and vaccine statements were detected either at the postgraduate (Honors and above) or undergraduate (BSc and MBBS) level. Moreover, the level of positive vaccine sentiments expressed by both levels of education was very high, in all cases >90%, as shown in Appendix A. This table also shows that individuals with either level of education were equally persuaded that vaccines are compatible with their religious beliefs (*p* = 1.000). The positive sentiment expressed for the intention to receive a COVID-19 vaccine when one becomes available was slightly lower than those expressed for the vaccine confidence statements; it was, nevertheless, high, at >89%. No association was found between levels of education and the intention to receive a COVID-19 vaccine when one becomes available (*p* = 1.000).

The re-categorization of levels of education to Honors degree or lower versus Master’s and above led to a significant association between the level of education and the belief that vaccines are important for the respondent to have (*p* = 0.043). This is shown in Appendix A. A slightly lower percentage of those with up to Honors education agreed that vaccines were important for them to have, as compared with those with a Master’s degree and higher (94.5% vs. 97.6%); however, this small difference might not be practically important. A high level of positive vaccine sentiment was expressed by both levels of education, with >90% agreement with the all vaccine confidence statements. However, a slightly lesser proportion of the up to Honors group (88.4%) indicated their willingness to take a COVID-19 vaccine when one becomes available, compared to 92.0% of the Master’s and above group, but this difference was not statistically significant. These data are also presented in Appendix A.

## 4. Discussion

In this study, the vaccine confidence levels of healthcare workers in training and staff of the named training institution were estimated.

Our results show high levels of positive vaccine sentiment expressed by the study respondents across and within all groups investigated. This implies high levels of confidence in the vaccine constructs investigated, which were as follows: importance for children and self, safety, effectiveness, and compatibility with religious beliefs. The intention to receive a COVID-19 vaccine when one becomes available was also investigated. Although vaccination intention does not always translate to action [30,31], nevertheless, the positive disposition of current and future healthcare workers of the study population is potentially indicative of the better promotion of the uptake of routine and COVID-19 vaccines and vaccination. This is assumed because of the important influence of healthcare workers as a credible source of information for the public on routine childhood vaccines [14,15,16] and, more recently, COVID-19 vaccines [32]. Our results are also consistent with the findings of a similar study that reported on the vaccine sentiments of the more educated, urban-dwelling online adult population portion of the general South African public [33]. While this 2020 survey reported that 64% of their described study population were willing to receive a COVID-19 vaccine when one becomes available, our results report a higher 89.5%. This could be due to the relatively homogenous nature of our study population, and the more recent timing of our study. The findings of our study are congruent with those of Lazarus et al. [31], who reported that 81.58% of the South African population included in their study indicated their willingness to receive a COVID-19 vaccine, and further supported by the results of another recent study, which found that an estimated 79% of Africans are willing to receive a COVID-19 vaccine [34]. Results from surveys conducted in other parts of the world also report high levels of willingness to receive a COVID-19 vaccine [31,32,35,36].

The high level of intention to receive a COVID-19 vaccine reported in our study was, however, comparatively less than that reported for the five vaccine confidence statements, all of which were ≥95%. One reason that we suggest for this is that COVID-19, being a relatively new disease, does not yet have its medium- and long-term prognosis well understood and defined, as with other vaccine preventable diseases (VPDs) for which vaccines are routinely administered, and that the vaccines are newly developed. However, this may not be the only reason for this observed trend.

A log binomial regression of the intention to receive a COVID-19 vaccine when one becomes available (the outcome of interest) against the five vaccine statements revealed confidence in the safety and effectiveness of such vaccines as the most likely predictors of the intention to receive it. This finding is noteworthy as safety and, to a lesser degree, effectiveness concerns have been cited in the literature as some of the leading reasons for reluctance to receive a COVID-19 vaccine [30,32,37,38]. In seeming contradiction, the consideration of the safety and effectiveness of a COVID-19 vaccine has also been proffered as a reason to receive the vaccine [31,39,40,41,42]. These two constructs are pivotal to the acceptance or otherwise of any vaccine, and are often included in the items used to query the intention to receive a COVID-19 vaccine, as exemplified by the study of Lazarus et al. [31] and the Africa Centre for Disease Control and Prevention (Africa CDC) survey [34].

With particular regard to COVID-19 vaccines, caregivers were reported to be more willing to vaccinate their charges than themselves [39], while the opposite was reported about healthcare workers as parents [43]. Therefore, the high levels of positive sentiments expressed by our study participants, particularly for the importance of vaccines for self and children, could be considered a potential indication of better vaccine uptake and coverage in the population that will be served by these healthcare workers in the future.

Religious beliefs can have a positive or negative influence on vaccination activities, as previously documented in the literature [44,45,46,47,48]. In our study population, respondents in the various religious groups investigated showed high levels of positive vaccine sentiments, with the exception of the one highlighted earlier. Respondents who considered vaccines to be compatible with their religious belief were twice as likely to indicate their willingness to receive a COVID-19 vaccine when one becomes available.

There was marginal variation in the level of the other demographic factors, with most showing consistently high levels of agreement with the five vaccine confidence statements and a slight decrease in the intention to receive a COVID-19 vaccine, as shown in the Appendix A. Staff members showed slightly higher levels of agreement with three out of the five vaccine confidence statements and the intention to receive a COVID-19 vaccine than students or the “both staff and student” group; they had equal levels of agreement with the students with the vaccine efficacy statement, which was minimally higher than that of the “both staff and student” group. Only in the “vaccine is compatible with my religious belief” statement did the student group have a slightly higher agreement level than the staff and “both” groups (96.7% versus 94.8% and 94.9%, respectively). This variation in the levels of agreement (indicative of levels of confidence), though minimal and not statistically significant, is, nevertheless, important. It shows that staff members having higher levels of vaccine confidence are rightly positioned to influence and impact future healthcare workers while they are still in training. This further reinforces the anticipation that future healthcare workers will have positive vaccine sentiments, which, in turn, should translate to better vaccine recommendation to and vaccine uptake in the population served by such healthcare workers [16,37,49].

The male respondents in our study showed marginally but not statistically significantly higher levels of agreement with three out of the five vaccine statements, the exceptions to this trend being the “vaccine are important for children to have” and the “vaccines are compatible with my religious belief” statements. The reasons for the exceptions could be that females are generally the primary caregivers to children (especially infants and pre-teens), and, as anecdotal evidence suggests, females are more inclined to religious activities than males. Nevertheless, this finding aligns with the findings of some recently published studies [43,50,51,52], but is in contrast with the report in the Kaiser Health News [53], which found that females were more willing than males to receive a COVID-19 vaccine in the United States of America.

The age and number of years of post-high school schooling also follows a similar trend to those of the other demographic factors. The participants who agreed with the five vaccine statements and the intention to receive a COVID-19 vaccine when one becomes available had a median of 6.0 number of years of post-high school schooling. In contrast, the median years of those who disagreed with the vaccine confidence statements and intention to receive a COVID-19 vaccine oscillated between 5.0 and 5.5. The median age distribution in years for the participants that agreed with the five vaccine confidence statements was consistently 29, and it was 30 for those who indicated their willingness to receive a COVID-19 vaccine. On the contrary, the median age for those who disagreed varied considerably, from 24 to 33, as shown in the Appendix A. Nonetheless, it was also 30 years for those who indicated their unwillingness to receive a COVID-19 vaccine.

This trend changed when the ages were grouped together. The age group ≤24 years showed a lesser degree of agreement with three out of the five vaccine confidence statements; only in the “vaccines are important for children to have” and “vaccines are compatible with my religious beliefs” did they have a slightly higher levels of agreements than the ≥65 year olds. These, on the other hand, were more likely to indicate their willingness to receive a COVID-19 vaccine when one becomes available. This finding is also supported by the recent literature [37,50,51,54].

Our study found no significant association between level of education (undergraduates and post graduates) and four of the five vaccine confidence statements. These findings are consistent with previous studies that found higher levels of education among healthcare workers to be associated with greater vaccine confidence [37,49,55]. The level of education in our study population can be considered to be relatively high, as most respondents had at least some form of post-high school schooling. Therefore, the findings of our study should be viewed with this in mind.

This study is not without limitations. The low response rate characteristic of online surveys is one such limitation. However, the actual response rate of 21.79%, which gave an actual sample size of 1015, was sufficient, providing adequate precision for the study as it was within the estimated sample size range calculated in the published protocol [20]. The study population was similar to the target population, using gender distribution as an indicator. Nevertheless, it should be noted that the views expressed by the study sample population may not adequately represent the views of those not supportive of vaccination in the target population. Thus, the possibility of selection bias cannot be ruled out due to the voluntary nature of the survey. No diploma and not all possible degrees were included in the response options for the question “highest degree obtained” and this may have had some influence on the response to the question.

## 5. Conclusions

We conclude that the levels of vaccine confidence and the intention to receive a COVID-19 vaccine when one becomes available in our study population were very high in the immediate period prior to the initiation of the vaccine programme in South Africa. The implication of these findings is that, despite all the debates and uncertainties that were rampant during this period, healthcare workers and students in our study population had highly positive vaccine sentiments, as expressed by their high vaccine confidence levels and the intention to receive a COVID-19 vaccine when one becomes available, though it remains to be seen if the expressed intention translates to corresponding action. Nevertheless, more emphasis on vaccine education and promotion will be beneficial among these current and future healthcare workers. This will potentially further enhance vaccine acceptance and uptake in the general population in the short, medium, and long term.

The novelty of our study lies in the fact that, to the best of our knowledge, no such study had been carried out in the study population or in a similar context in South Africa at such a critical period in time (that is, just prior to the COVID-19 vaccination roll-out, in which healthcare workers were among the first group scheduled for vaccination).

## Figures and Tables

**Table 1 vaccines-09-01246-t001:** Characteristics of the study sample population.

Demographic Variables		Count	Percentage
Staff/Student	Staff	257	25.3
	Student	675	66.5
	Both	83	8.2
Sex	Male	255	25.1
	Female	758	74.7
	Other	2	0.02
Degree	BSc	370	36.5
	Hons	135	13.3
	MBBS	219	21.6
	MSc	200	19.7
	PhD	91	9.0
Religion	Islam	111	11.0
	Roman Catholic	91	9.0
	Orthodox	23	2.2
	Pentecostal	155	15.0
	Traditional	75	7.4
	Jewish	8	0.8
	Buddhist	4	0.4
	Hindu	25	2.5
	Atheist	76	7.5
	Agnostic	73	7.2
	Other	365	36.0
	7th Day Adventist	9	1.0
Post-matriculation years of schooling	≤5	449	44.2
	6–10	283	27.9
	11–14	147	14.5
	15–20	52	5.1
	≥21	84	8.3
Age group	≤24	414	40.8
	25–34	243	23.9
	35–44	193	19.0
	45–54	84	8.3
	55–64	70	6.9
	≥65	11	1.1

The total number of valid responses was 1015 for all categories. Values given are absolute counts and percentages for each variable.

**Table 2 vaccines-09-01246-t002:** Vaccine sentiments among the study population.

Statements	Strongly Agree	Tend to Agree	Don’t Know	Tend to Disagree	Strongly Disagree
1. Vaccines are important for children to have	862 (84.93%)	107 (10.54%)	20 (1.97%)	19 (1.87%)	7 (0.69%)
2. Vaccines are important for me to have	760 (74.88%)	180 (17.73%)	30 (2.96%)	32 (3.15%)	13 (1.28%)
3. Overall, I think vaccines are safe	560 (55.17%)	335 (33.00%)	77 (7.59%)	31 (3.05%)	12 (1.18%)
4. Overall, I think vaccines are effective	603 (59.41%)	338 (33.30%)	49 (4.83%)	13 (1.28%)	12 (1.18%)
5. Vaccines are compatible with my religious beliefs	704 (69.36%)	180 (17.73%)	95 (9.36%)	19 (1.87%)	17 (1.67%)
6. I will take a COVID-19 vaccine when one becomes available	631 (62.17%)	150 (14.78%)	142 (13.99%)	46 (4.53%)	46 (4.53%)

The total number of valid responses was 1015 for all statements. Values given are absolute counts (percentages) for each response.

**Table 3 vaccines-09-01246-t003:** The percentage of agreement with the vaccine confidence statements and intention to receive a COVID-19 vaccine.

Statements	Agree	Confidence Interval for Agreement
1. Vaccines are important for children to have	97.4% (*n* = 969)	96.14–98.25
2. Vaccines are important for me to have	95.4% (*n* = 940)	93.88–96.61
3. Overall, I think vaccines are safe	95.4% (*n* = 895)	93.82–96.62
4. Overall, I think vaccines are effective	97.4% (*n* = 941)	96.15–98.28
5. Vaccines are compatible with my religious beliefs	96.1% (*n* = 884)	94.57–97.21
6. I will take a COVID-19 vaccine when one becomes available	89.5% (*n* = 781)	87.19–91.38

**Table 4 vaccines-09-01246-t004:** Association between vaccine confidence statements and intention to receive a COVID-19 vaccine.

Vaccine Confidence Statements	Vaccine Sentiments	Agree	Disagree	Total	Crude Relative Risk (95% CI)	*p*-Value
1. Vaccines are important for children to have	Agree	91.7% (*n* = 767)	8.3% (*n* = 69)	100% (*n* = 836)	3.5 (1.78–6.99)	<0.001
	Disagree	26.1% (*n* = 6)	73.9% (*n* = 17)	100% (*n* = 23)		
	Total	90% (*n* = 773)	10% (*n* = 86)	100% (*n* = 859)		
2. Vaccines are important for me to have	Agree	94.6% (*n* = 772)	5.4% (*n* = 44)	100% (*n* = 816)	18.5 (4.78–71.12)	<0.001
	Disagree	5.1% (*n* = 2)	94.9% (*n* = 37)	100% (*n* = 39)		
	Total	90.5% (*n* = 774)	9.5% (*n* = 81)	100% (*n* = 855)		
3. Overall, I think vaccines are safe	Agree	94.7% (*n* = 749)	5.3% (*n* = 42)	100% (*n* = 791)	32.2 (4.67–221.89)	<0.001
	Disagree	2.9% (*n* = 1)	97.1% (*n* = 33)	100% (*n* = 34)		
	Total	90.9% (*n* = 750)	9.1% (*n* = 75)	100% (*n* = 825)		
4. Overall, I think vaccines are effective	Agree	93.3% (*n* = 762)	6.7% (*n* = 55)	100% (*n* = 817)	21.4 (3.16–145.82)	<0.001
	Disagree	4.3% (*n* = 1)	95.7% (*n* = 22)	100% (*n* = 23)		
	Total	90.8% (*n* = 763)	9.2% (*n* = 77)	100% (*n* = 840)		
5. Vaccines are compatible with my religious beliefs	Agree	93.2% (*n* = 715)	6.8% (*n* = 52)	100% (*n* = 767)	2.2 (1.46–3.78)	<0.001
	Disagree	41.4% (*n* = 12)	58.6% (*n* = 17)	100% (*n* = 29)		
	Total	91.3% (*n* = 727)	8.7% (*n* = 69)	100% (*n* = 796)		

## Data Availability

All data relevant to the study are included in the article or uploaded as Appendix A.

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
