# Peer review of "Estimating Vaccine Confidence Levels among Healthcare Staff and Students of a Tertiary Institution in South Africa"

_vaccines, 2021, doi:10.3390/vaccines9111246_

Round 1

Reviewer 1 Report

In the manuscript titled “Estimating vaccine confidence levels among healthcare staff and students of a tertiary institution in South Africa” by Wiysonge et.al., the author did a systematic survey-based study among healthcare staff and students at a tertiary institution in South Africa to elucidate their vaccine confidence level based on the religious belief, sex, education etc. The authors concluded their finding by mentioning the vaccination confidence among the study population is high. This internet-based survey enabled the author to support their conclusion. The authors described how the socio demographic and religious belief influenced the vaccine confidence.

There might me limitation of the survey-based study and this study was done around February and March, this reviewer belief this manuscript is suitable for Vaccines.

Author Response

The authors thankfully acknowledge the efforts of Reviewer 1 in reviewing the manuscript, and gratefully receives the comments.

The study limitations are reported by the authors in the last paragraph of the discussion. The data collection was conducted between February and March 2021. This was the period just before the commencement of the COVID-19 vaccination roll-out in South Africa in which Healthcare workers were the first group to receive the vaccines. Hence, this was judged to be the best period to estimate their vaccine confidence and their intention to receive a COVID-19 vaccine.  

Reviewer 2 Report

This is an interesting paper on attitudes toward vaccine acceptance among a sample of health workers and students in South Africa. The topic is important, as Health workers are the most trustfull source of information and as they come into contact with the disease, were required in many countries to be the first in line in taken the vaccine. The paper is based on an online survey, is generally well written and provide interesting results.

Comments:

  1. Introduction, I found very odd the use of the word "infamous appearance of the coronavirus". it is a pandemic, and nothing is infamous to describe the virus (another virus out of millions) or the pandemic.
  2. The introduction clearly identifies the motives for the study and its importance. However, it lacks conceptualization as it does not direct us to what is the framework of though of what are the variables that may affect health workers confidence. Furthermore, it does not indicate how this study contributes to the existing literature and what is new in this study that we did not know before.
  3. I am not sure that a reader will understand what is an occassional student. Can the authors explain the concept?
  4. The item describing the dependent variable, " 1 intention to receive a COVID-19 vaccine when one becomes avail- 
    able statement." is not presented in the methods part. Can you show the exact wording of the dependent variable and the response categories?
  5. Author/s mention high levels of multicollinearity. Can you explain more in detail the statement (multicollenarity of which variables and what was the VIF)
  6.  

Author Response

Response to Reviewer 2 Comments

The authors use this medium to thank this Reviewer for the efforts at reviewing our manuscript and the constructive feedback given to help to further improve the quality of the manuscript.

Please find below detailed response to each point raised.

Point 1: Introduction, I found very odd the use of the word "infamous appearance of the coronavirus". it is a pandemic, and nothing is infamous to describe the virus (another virus out of millions) or the pandemic.

Response 1: The word “infamous” has been removed from the manuscript as recommended.

Point 2: The introduction clearly identifies the motives for the study and its importance. However, it lacks conceptualization as it does not direct us to what is the framework of though of what are the variables that may affect health workers confidence. Furthermore, it does not indicate how this study contributes to the existing literature and what is new in this study that we did not know before.

Response 2: This study builds on the study of Larson and her team (2016) (reference 28), an international study that examined the perceptions of vaccine importance, safety, effectiveness, and religious compatibility among 65,819 individuals across 67 countries. The framework of the 67-country study (and by extension, our study) was based on the vaccine confidence index developed by Larson and her team to map confidence in vaccines and immunization programmes worldwide. This is now indicated in the Introduction section. 

The variables that may affect healthcare workers confidence include all the demographic variables investigated such as age, gender, level of education (as reflected by post-matriculation year of schooling and degree held) and religion. This is now clearly stated in the Methods section 2.4 (Data collection) of the manuscript.

The current study expands on, and provides more recent data on a subset of the population of one of the 67 countries included in the 2016 study. Furthermore, what this study contributes to existing literature is explored in detail in the Discussion section. Here the findings of the study were appropriately situated among existing literature. Moreover, its novel finding (that is, the high levels of vaccine confidence among healthcare workers and students, and the high level of willingness to receive a COVID-19 vaccine when one becomes available) has not been reported among this study population; especially in the period just prior to the COVID-19 vaccination roll-out when there was so much debate and uncertainties surrounding the COVID-19 vaccines. This was true to the best of our knowledge at the conceptualization of this study.

We have now elaborated on these issues and included a brief synopsis of the novelty and implication of the findings in the Conclusion section of the revised manuscript.

Point 3: I am not sure that a reader will understand what is an occassional student. Can the authors explain the concept?

Response 3: Occasional students are students that are taking modules for non-degree purposes at the university.

This explanation is now included in the Methods section 2.2. Under “Study population description and sampling strategy” 

Point 4: The item describing the dependent variable, " 1 intention to receive a COVID-19 vaccine when one becomes avail-able statement." is not presented in the methods part. Can you show the exact wording of the dependent variable and the response categories?

Response 4: The item describing the dependent variable, "intention to receive a COVID-19 vaccine when one becomes available" was presented in the methods part. The exact wording as used in the Methods section 2.4 (Data collection) is: “I will take a COVID-19 vaccine when one becomes available”.  This is also the exact wording used in the questionnaire and in the Results section, in Table 2 titled: Vaccine sentiments among the study population (statement 6). This exact wording is now included in section 2.4 where the statement for this comment was extracted, it is repeated in the last paragraph of section 2.4 where the response categories are stated, and also in section 2.5 (Data analysis). The initial appearance of the exact wording of the text has now been highlighted in red for greater visibility to the reviewer.

Point 5: Author/s mention high levels of multicollinearity. Can you explain more in detail the statement (multicollenarity of which variables and what was the VIF)

Response 5: All five independent variables (the 5 vaccine confidence statements) were highly associated with each other. These were expressed as binary variables (agree/disagree). It is not possible to calculate VIF (variance inflation factor) between categorical or binary variables. Multicollinearity in such cases is normally shown with chi square tests.  Chi square tests between each pair of independent variables (1 to 5) showed a highly statistically significant association in each case (p<0.001 in all pairwise cross-tabulations) leading us to conclude that high levels of consistency exists in the answers to each of the 5 statements and thus high levels of multicollinearity. It was therefore not possible to put all 5 statements in a model together as independent variables.

A condensed version of this detailed explanation has now been included in the revised manuscript. It is briefly introduced in the last paragraph of the Methods section 2.5 (Data analysis), but described in details in the Results section between Table 1 and Table 2.

Round 2

Reviewer 2 Report

the author/s had addressed my previous concerns